# Regional Research-Practice-Policy Partnerships in Response to Climate-Related Disparities: Promoting Health Equity in the Pacific

**DOI:** 10.3390/ijerph19159758

**Published:** 2022-08-08

**Authors:** Lawrence A. Palinkas, Meaghan O’Donnell, Susan Kemp, Jemaima Tiatia, Yvonette Duque, Michael Spencer, Rupa Basu, Kristine Idda Del Rosario, Kristin Diemer, Bonifacio Doma, David Forbes, Kari Gibson, Joshua Graff-Zivin, Bruce M. Harris, Nicola Hawley, Jill Johnston, Fay Lauraya, Nora Elizabeth F. Maniquiz, Jay Marlowe, Gordon C. McCord, Imogen Nicholls, Smitha Rao, Angela Kim Saunders, Salvatore Sortino, Benjamin Springgate, David Takeuchi, Janette Ugsang, Vivien Villaverde, Kenneth B. Wells, Marleen Wong

**Affiliations:** 1Suzanne Dworak-Peck School of Social Work, University of Southern California, Los Angeles, CA 90089, USA; 2Phoenix Australia, Centre for Posttraumatic Mental Health, Department of Psychiatry, University of Melbourne, Melbourne, VIC 3010, Australia; 3School of Counseling, Human Services and Social Work, University of Auckland, Auckland 1010, New Zealand; 4Te Wānanga o Waipapa, School of Māori Studies and Pacific Studies at the University of Auckland, Auckland 1010, New Zealand; 5Asian Disaster Preparedness Center, Bangkok 10400, Thailand; 6School of Social Work, University of Washington, Seattle, WA 98195, USA; 7Office of Environmental Health Hazard Assessment (OEHHA), California Environmental Protection Agency, Sacramento, CA 95812, USA; 8Office of the President, University of Nueva Caceres, Naga 4400, Philippines; 9School of Social Work, University of Melbourne, Melbourne, VIC 3010, Australia; 10Department of Chemical Engineering, Mapua University, Manila 1102, Philippines; 11School of Global Policy and Strategy, University of California, San Diego, CA 92093, USA; 12Provincial Government of New Ireland, Kavieng 631, Papua New Guinea; 13Department of Epidemiology and Chronic Disease, School of Public Health, Yale University, New Haven, CT 06520, USA; 14Department of Population and Public Health Sciences, Keck School of Medicine, University of Southern California, Los Angeles, CA 90007, USA; 15International Organization for Migration, Canberra, ACT 2601, Australia; 16College of Social Work, Ohio State University, Columbus, OH 43210, USA; 17International Organization for Migration, Majuro 96960, Marshall Islands; 18School of Medicine, LSU Health Sciences Center—New Orleans, School of Medicine and School of Public Health, New Orleans, LA 70112, USA; 19School of Public Health, LSU Health Sciences Center—New Orleans, School of Medicine and School of Public Health, New Orleans, LA 70112, USA; 20Center for Health Services and Society, Jane and Terry Semel Institute, Department of Psychiatry and Biobehavioral Sciences, David Geffen School of Medicine, University of California, Los Angeles, CA 90095, USA; 21Department of Health Policy and Management, Fielding School of Public Health, University of California, Los Angeles, CA 90095, USA

**Keywords:** health equity, Pacific region, climate change, Small Island Developing States, low- and middle-income countries, disasters, social determinants of health

## Abstract

Although climate change poses a threat to health and well-being globally, a regional approach to addressing climate-related health equity may be more suitable, appropriate, and appealing to under-resourced communities and countries. In support of this argument, this commentary describes an approach by a network of researchers, practitioners, and policymakers dedicated to promoting climate-related health equity in Small Island Developing States and low- and middle-income countries in the Pacific. We identify three primary sets of needs related to developing a regional capacity to address physical and mental health disparities through research, training, and assistance in policy and practice implementation: (1) limited healthcare facilities and qualified medical and mental health providers; (2) addressing the social impacts related to the cooccurrence of natural hazards, disease outbreaks, and complex emergencies; and (3) building the response capacity and resilience to climate-related extreme weather events and natural hazards.

## 1. Introduction

Climate change is a global phenomenon requiring a global response, especially with respect to the detrimental consequences to health, which have been long ignored [1,2]. However, a regional approach may be more suitable, appropriate, and appealing than a global approach. Unique features of geography, localized climate change impacts, population, priorities, and preferences in adapting to climate change, and contextual vulnerabilities align with a regional approach to building a capacity to address climate change-related physical and mental health disparities [2]. For instance, low- and middle-income countries (LMICs) are especially vulnerable to adverse health consequences because of their location and limited resources to address climate-related threats [1,2]. Although they are not the largest contributors to climate change due to increased greenhouse gas emission production, they are the populations most affected by it. High-income countries (HICs) can play an important role in ensuring that LMICs everywhere have the resources to respond to these challenges [3], but a regional approach requires partnerships between and among HIC and LMIC researchers, practitioners, and policymakers and LMIC community stakeholders.

In this commentary, we describe one such regional approach comprised of partnerships involving researchers, practitioners, and policymakers located along and within the Pacific Rim dedicated to addressing the health inequities that are either a consequence of climate change or have been exacerbated by climate change in the Pacific Region. With the overall goal of demonstrating the potential value of a regional approach, the aims of this commentary are threefold: (1) to describe the impacts of climate change on health equity in the Pacific Region, (2) to outline the priorities for addressing these impacts on health equity, and (3) to describe the efforts of the PAcific RIm Climate Health Equity (PARICHE) network to conduct research and capacity building in accordance with these priorities.

## 2. Climate Health Equity in the Pacific Region

One example of the need for such an approach are the Small Island Developing States (SIDS) and territories and larger LMICs within the Pacific Region that are disproportionately impacted by climate change relative to other regions in the world. In this region, climate change has potentially devastating implications for Indigenous peoples and their traditional ways of life [2,4] (Figure 1). Island nations such as Vanuatu, Tuvalu, Kiribati, and the Marshall Islands are projected to experience the largest relative increase in flood risk due to the sea level rise (SLR) in this century [2]. Atoll countries (SIDS located on ring-shaped coral reefs, islands, or series of islets, such as Kiribati, the Marshall Islands, Nauru, Tuvalu, and Tokelau) are particularly susceptible to water insecurity from climate change, being dependent on rainwater or freshwater aquifers [5]. Increasing ocean acidification is also likely to impact coral reefs and coastal ecosystems, potentially leading to a reduction in local food stocks and marine resources, greater costs for food and water, and reductions in tourism and household incomes [6]. Entire communities have already been forced to relocate because of persistent flooding.

Flooding and high winds from tropical cyclones have also contributed to displacement in the region. About 4000 residents of Tuvalu and 3300 residents of Vanuatu were displaced by Cyclone Pam in 2015 [8]. In 2013, Typhoon Haiyan resulted in 6300 deaths and 4.1 million people displaced [9]. The high cost of adapting to the sea level rise and flooding is a powerful driver of migration within or from SIDS in the Pacific, resulting in resettlement on marginal land prone to flooding or in increasingly crowded urban centers and significant impacts on cultural traditions and mental health [10].

Projected changes in the physical environment will have significant impacts on human health (e.g., changing disease vectors, chronic diseases, adverse birth outcomes, and water-borne diseases) [11]. The most direct effect will be the further reduction of already declining agricultural output per capita [12]. Malnutrition, a common challenge for children in many Pacific SIDS, is linked with communicable disease risk and has profound implications for child health and development [11].

Another health-related consequence of climate change is the physical and mental trauma associated with extreme weather events such as cyclones. The mortality and injury rates are disproportionately concentrated in less-developed nations of the Western Pacific. Storm surges and floods are the primary causes of death in cyclones [13]. The cyclone mortality risk is particularly high in lower-income countries, with lacerations, wounds, contusions, blunt trauma, animal/insect bites, and motor vehicle injuries among the most frequent types of injuries reported [13]. Mental health problems are also a consequence of extreme weather events. Elevated levels of depression, anxiety, substance abuse, suicides, homicides, violent behavior, and post-traumatic stress disorder following major high-impact weather events are common throughout the Pacific Region, may be disabling, and can complicate recovery efforts for large segments of affected populations [14]. The widespread effects of SLR on individuals and their families, including fear and worry on personal and community levels, and the loss of ancestral lands and traditional livelihoods, have also been reported [14]. However, cultural barriers, social discrimination, and prejudice regarding mental health-related challenges often prevent residents of SIDSs from seeking treatment or resolving underlying issues [14]. As such, culturally appropriate and acceptable mental health services will be critical [15].

Climate change is also likely to be an important multiplier of the existing social and health problems experienced by Pacific populations. The health status of Pacific Island communities is generally poor, with average life expectancies 10 years or more below those of HICs and rates of obesity and noncommunicable diseases among the highest in the world [11,16]. Increased exposure to warm temperatures will significantly increase the mortality and morbidity risk of individuals with chronic conditions such as cardiovascular disease, diabetes, and kidney disease [13]. Taken together, climate change will exacerbate disparities in health, particularly for populations with lower socioeconomic status and more chronic stress throughout the region [15,17].

## 3. A Regional Solution to Climate Health Equity in the Pacific

Although many of these impacts of climate change on health equity are not unique to the Pacific Region, a regional approach to addressing them is recommended for three reasons. First, even though the problems related to climate change are global in nature, responses are local and based on resource availability; prior experience; and cultural systems of knowledge, attitudes, and behaviors. For instance, many of the same impacts of climate change on well-being and the factors that mitigate these impacts can be found in other parts of the world, such as Wales [18]. Like Wales, the Pacific Region is confronting increased food insecurity and SLR. However, unlike Wales, SLR in the Pacific is threatening to immerse entire nation states on low-lying islands, and Pacific SIDS have fewer resources to develop the infrastructure necessary to mitigate the impacts of SLR. Moreover, community responses to dealing with food insecurity in the Pacific Region (e.g., traditional kin-based networks of food exchange and social support) are very different from those in operation in Wales (government assistance and nongovernment organization food distribution programs). Even within the Pacific Region, there is a wide diversity of experiences, perspectives, and emotional responses to climate change [19].

Second, global approaches tend to ignore or underappreciate Indigenous perspectives on solving these problems. Anthropogenic climate change is intimately connected to the ideologies, systems, and practices of colonialism, and the impacts on Indigenous peoples can be conceptualized as an intensification of the process of colonization [20]. In the Pacific Region, efforts to address the impacts of climate change have given rise to calls for a “Pan-Pacific identity”, solidarity, and self-determination [21,22].

Third, past experiences of implementing evidence-based policies, practices, and interventions suggests that outcomes of these efforts are much less likely to be successful and sustainable unless the intended beneficiaries are engaged at the very beginning of the process of translating research to policy or practice [23,24,25]. This is especially true in LMICs where the power differentials between well-resourced researchers and consultants from HICs and their less-resourced counterparts lead to resistance in adopting “foreign” solutions to local problems [26].

The PAcific Rim Climate and Health Equity Network (PARICHE), a network of researchers, practitioners, and policymakers formed with the aim of responding to the growing health and mental health disparities in the Pacific, provides a potential exemplar of a regional approach to climate health equity. The mission of PARICHE is to address the health disparities associated with global climate change in the Pacific Region by building the capacity for translational research in SIDS, other LMICs, and Indigenous communities in HICs. At present, the network includes representatives from academic institutions in the United States, Australia, New Zealand, and the Philippines and representatives from SIDS/LMIC government agencies (e.g., Provincial Government of New Ireland, Papua New Guinea) and international organizations (e.g., Asian Disaster Preparedness Center, International Organization for Migration—Marshall Islands, Federated States of Micronesia, and Western Pacific) based in the region. Our approach to addressing climate and health equity involves establishing closer community–academic ties through a community-partnered participatory research (CPPR) approach [27] to collaboration. A form of community-based participatory research [23], partners are valued equally and collaborate jointly in research development, implementation, and dissemination; maintain respectful engagement across diverse ways of viewing and gaining knowledge; and anticipate and embrace the struggles and conflicts inherent in balancing diverse perspectives.

In the Pacific Region, the current priorities are focused on the capacity to respond to climate-related extreme weather events and, most recently, the COVID-19 pandemic. Limited healthcare facilities—especially hospitals—and qualified medical providers are perhaps the most important health issues facing the region today. The primary health issues in several parts of the region are childhood vaccinations, malaria, tuberculosis, nutritional disorders, and childhood infectious diseases such as yaws. Pacific Islanders have a unique health profile that may put them at risk for preterm birth: they have a disproportionately high prevalence of obesity and related noncommunicable diseases compared with other populations [16]. The importance of responding to mental health challenges related to climate change has also been recognized throughout the region, evidenced by studies of mental health after typhoons [28] and distress related to climate change in general [19,29]. Also vitally important are interventions grounded in Indigenous and local knowledge systems and responsive to priorities in these communities [30].

A second regional priority is addressing the social impacts related to the co-occurrence of natural hazards, disease outbreaks, and complex emergencies. For instance, typhoons have had a profound impact on family support systems throughout the region, threatening well-being and creating uncertainty over whether to remain and invest in the future or to move elsewhere [14]. Deaths of parents and other family members have placed a greater emphasis on intergenerational trauma and post-disaster child protection. Children who lose their parents and other family members who serve as caregivers in such events face severely diminished educational opportunities, placing them in a persistent cycle of poverty [31]. Impacts such as these have economic consequences, leading to further disparities.

A third regional priority is the need to build a response capacity and resilience to climate-related extreme weather events and natural hazards. This includes the need to bolster the preparedness and resilience of vulnerable populations with the support of diverse partners and the need to train local government units on emergency capacity. This is particularly important in SIDS where populations are exposed to compound trauma, and the ability to respond emotionally to one extreme event is impacted by experiencing another adverse/traumatic event. Psychosocial interventions can help survivors recover from a traumatic event and build resilience to deal with other impactful events. There is also a need to ensure access to healthy, culturally relevant foods and sustainable agriculture resistant to climate change and reduce the risk of malnutrition, which is related to both infectious and noninfectious diseases [32]. Over avenues for the promotion of general well-being in the region include engagement in Indigenous “Pasifika arts [19], planned relocation of residents from SIDS at risk of SLR [33], and partnerships for the implementation of community-based health promotion programs [34].

To address these needs, PARICHE is dedicated to improving the capacity for delivering healthcare and reducing health disparities related to climate change by partnering with Indigenous communities throughout the region to conduct research and train local practitioners and policymakers in implementation science and disaster preparedness. One of our earliest efforts occurred in 2013 when a group of students and faculty from the University of Southern California School of Social Work, led by PARICHE members Marleen Wong and Vivien Villaverde, traveled to the Philippines in the aftermath of Typhoon Haiyan. The USC Humanitarian Mission to the Philippines was inspired by the traditional Philippine concept of Bayanihan, the Tagalog/Filipino word for “collective spirit” [35]. A two-day training workshop for representatives of nongovernment organizations, private foundations, and government agencies was held with the goal of building participants’ knowledge and skills in the recovery and community rebuilding process. One year later, in partnership with the Republic of the Philippines Department of Education, Wong and Villaverde conducted training for the northern region in Tagaytay and the southern region in Cebu, focused on building a capacity for crisis interventions in schools. The 66 attendees from all over the country included teachers, counselors, nurses, school crisis responders, and administrators. The training had three components: (1) an overview of disaster response and recovery; (2) an outline of the impacts of disaster on human environment and the use of the three-tiered model for disaster preparedness planning [36]; and (3) the use of evidenced-informed interventions in schools (Psychological First Aid-Listen, Protect, Connect [37], and Healing After Trauma Skills [38]). The goal of the endeavor was to increase the knowledge and skills capacity for disaster response planning for implementation at the attendee’s respective sites. The participants developed the following action plans: (1) use the interventions (PFA-LPC and HATS) and connect with local administrators to plan division training, (2) inventory the current response partners and fill in the gaps with new partnerships, (3) develop division memos guided by the three-tiered model for disaster preparedness planning to support and keep the work moving, and (4) support the local regions to build a monitoring and evaluation process. This effort led to the development of a school-based infrastructure for disaster response throughout the Philippines.

A more recent example of the work conducted by network members was a collaboration in 2018 with Tuvalu Association of Non-Government Organisations (TANGO) and the Congregational Christian Church of Tuvalu in the development and testing of a low-intensity mental health intervention specifically designed to support emotional recovery after exposure to natural hazards and promote resilience for future natural hazards [39], illustrated in Box 1.

Box 1Skills of life adjustment and resilience in Tuvalu.PARICHE members Kari Gibson, David Forbes, and Meaghan O’Donnell tested the feasibility, acceptability, efficacy, and safety of a culturally adapted version of Skills for Life Adjustment and Resilience (SOLAR) in two remote, cyclone-affected communities in the Pacific Island nation of Tuvalu. SOLAR was developed in 2015 by an interna-tional roundtable of disaster and mental health experts that included PARICHE mem-bers Forbes, O’Donnell, and Ugsang. It is a brief, trauma-informed psychosocial pro-gram that aims to reduce the distress and impairment associated with disaster expo-sure [39]. SOLAR utilizes a task-shifting approach by training lay “coaches” to deliver the intervention and was developed to be adapted for varied geographic, so-cio-economic, and cultural populations. An initial pilot study of SOLAR delivered by local community workers after bushfires in Australia found it to be feasible, acceptable, and safe to deliver [39]. It also provided preliminary evidence that SOLAR was effica-cious in reducing post-traumatic stress, anxiety, and depressive symptoms, although the small sample size prevented firm conclusions from being drawn [40]. In Tuvalu, SOLAR was administered to a treatment group (*n* = 49) by local, non-specialist facilita-tors (i.e., “coaches”) in a massed, group format across 5 consecutive days. The control group (*n* = 50) had access to the usual care (UC). Large, statistically significant group differences in psychological distress were observed after controlling for the baseline scores in favor of the SOLAR group. The mean group outcomes were consistently lower at the 6-month follow-up than at the baseline. SOLAR was found to be acceptable and safe, and the program feedback from participants and coaches was overwhelmingly positive [39]. The feasibility of SOLAR in this setting provided confidence in its poten-tial rollout in other LMIC settings where mental health literacy is limited.

In collaboration with the providers of the Samoa National Health Services and the Samoa Ministry of Health, PARICHE member Nicola Hawley studied the social and environmental determinants of obesity [41] and services for chronic noncommunicable diseases in Samoa [42]. The aims and outcomes of this partnership are described in Box 2.

Box 2Chronic diseases in Samoa.Nicola Hawley and colleagues at Yale University are collaborating with healthcare providers in Samoa and American Samoa to understand how maternal and child health are impacted by rising levels of obesity and diabetes in resource-poor settings [43]; determine how innovations in healthcare delivery can impact the identification and treatment of obesity-related disease during the perinatal period; and develop inter-ventions focused on pregnancy, childhood, and adolescence to prevent the intergenera-tional transmission of obesity-related disease [44,45]. The Obesity, Lifestyle, And Ge-netic Adaptations (OLAGA; “life” in Samoan) study group uses a life course approach to understand the origins of obesity among Samoans and other Pacific Islanders and focuses on developing culturally relevant interventions to reduce the burden of obesity and obesity-related conditions. For instance, this team found a significant burden of diabetes and hypertension in Samoa, exceeding the recent prevalence estimates of other low- to middle-income countries by nearly two-fold, pointing to a severe unmet need in both detection and the subsequent control, and the monitoring of these chronic condi-tions exists [46]. Their results suggest that the initial diagnosis and surveillance stage in the cascade of care for chronic conditions should be a major focus of primary care ef-forts; national screening campaigns and programs that leverage village and district nurses to deliver community-based primary care may significantly impact the gap clo-sure in the NCD cascade.

PARICHE network members have also been engaged in assisting SIDS government agencies in disaster mitigation, relief, and reconstruction activities [33]. Box 3 provides an illustration of the International Organization for Migration Regional Office located in The Republic of the Marshall Islands, which has been engaged in assisting the Pacific Region in responding to the COVID-19 pandemic.

Box 3Responding to the COVID-19 pandemic in the Pacific.In 2020, the Republic of the Marshall Islands (RPI) conducted the first COVID-19 Tabletop Exercise (TTX) and Simulation among the Pacific Island nations. Attended by 296 participants, the event was organized by the International Organization for Migra-tion and World Health Organization in close collaboration with the RPI National Dis-aster Management Office under the Office of the Chief Secretary and the Ministry of Health and Human Resources. PARICHE network member Angela Saunders served as the lead facilitator and drill master for the event [47]. The TTX provided participants an opportunity to analyze, plan, and coordinate response strategies that their respective agencies would implement in the face of a COVID-19 outbreak during repatriation. The two-day exercise ended with the groups drafting action plans based on the gaps identi-fied during the TTX. One of the key outcomes of the event was the realization there was a need for continued outreach, inclusion, and practices with other key stakeholders such as traditional leadership and the private sector. The IOM will continue to support the participants in monitoring their action plans and continually assessing their pre-paredness for COVID-19.

PARICHE members have also been involved in working with Indigenous groups in HICs bordering the Pacific Region. Jemaima Tiatia has been working with the New Zealand Ministry of Pacific Peoples in developing the capacity to best support the mental health needs of climate change migrants [48]. Michael Spencer has worked with Native Hawai’ians in Hawai’i to provide evidence of the critical role culture and Indigenous knowledge can play in environmental justice policies and practices [30]. He is currently working with the Colville Indian Reservation in Washington State on the impact of climate change on foraging and hunting.

## 4. Policy Implications

In providing a regional perspective to the issue of climate change-related health equity, the PARICHE approach and experience offers several implications for the development and implementation of policies addressing climate change and health equity in other parts of the world. First, the principles of community-partnered participatory research are designed to eliminate the dominance of participants from HICs in climate health equity partnerships with participants from SIDS and LMICs. HICs have more resources but also bear more responsibility for human-induced climate change, while LMICs have fewer resources but are more exposed and more vulnerable to climate change impacts [26]. Second, the goal of the regional approach is to empower those most affected by climate change to control the process of addressing health equity by dictating priorities and identifying interventions and strategies for the implementation of policies, practices, and programs that are both feasible and acceptable. Third, local empowerment should be a key priority in the implementation of international incentives and coordinating for scaling up climate adaptation. Efforts to understand the means of effective climate risk reduction; assessing cascading, compounding, and transboundary risks; tracking adaptation progress; and financing adaptation such as those proposed by Magnan and colleagues [3] cannot succeed without simultaneously empowering communities in low-resource settings such as the SIDS and LMICs of the Pacific Region to engage in such activities as full partners.

## 5. Conclusions

PARICHE aims to fashion a forward-looking, community-centered framework that leverages scientific expertise coupled with lived knowledge systems in specific cultural and geographical contexts. Joining hands with frontline communities to devise climate action is fundamental to advancing social, economic, and environmental justice and addressing health disparities. Although the activities of PARICHE are informed by the specific impacts of climate change on health equity in the Pacific Region, it can serve as a model for the development of regional networks to address similar challenges across the globe.

## Figures and Tables

**Figure 1 ijerph-19-09758-f001:**
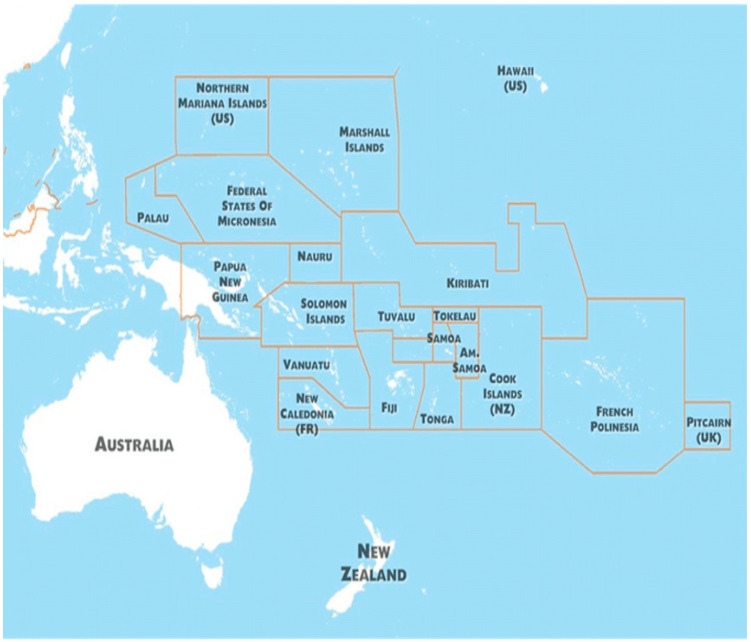
Countries and islands of the Pacific Region [7].

## Data Availability

Not applicable.

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
