# Peer review of "Regional Research-Practice-Policy Partnerships in Response to Climate-Related Disparities: Promoting Health Equity in the Pacific"

_ijerph, 2022, doi:10.3390/ijerph19159758_

Round 1
Reviewer 1 Report
This study developed to promoting the health equtity in the Pacific through regional research practice policy partnerships in response to climate related disparities. Findings represented a contribution to this area of literatures and were strengthened by this paper.
Substantial areas for clarification/consideration are noted below. The motivation of the study and policy implications need some further work. The paper would benefit from some further proofreading as there are types and grammatical errors. The authors may also make it clear in the text when describing the different trainers.
Detail comments
1. Please provide specific bullet pointed aims at the end of the introduction i.e., detail each research questions separately.
2. The authors should make the policy implications clearly about your study.
Author Response
Reviewer 1
This study developed to promoting the health equtity in the Pacific through regional research practice policy partnerships in response to climate related disparities. Findings represented a contribution to this area of literatures and were strengthened by this paper.
Substantial areas for clarification/consideration are noted below. The motivation of the study and policy implications need some further work. The paper would benefit from some further proofreading as there are types and grammatical errors. The authors may also make it clear in the text when describing the different trainers.
Detail comments
- Please provide specific bullet pointed aims at the end of the introduction i.e., detail each research questions separately.
Response: We have added the following at the end of the introduction (lines 75-82) to clarify the aims of the article.
“In this commentary we describe one such regional approach comprised of partnerships involving researchers, practitioners and policymakers located along and within the Pacific Rim dedicated to addressing the health inequities that are either a consequence of climate change or have been exacerbated by climate change in the Pacific Region. With the overall goal of demonstrating the potential value of a regional approach, the aims of this commentary are threefold: 1) to describe the impacts of climate change on health equity in the Pacific region; 2) to outline the priorities for addressing these impacts on health equity; and 3) to describe the efforts of the Pacific Rim Climate Health Equity (PARICHE) network to conduct research and capacity building in accordance with these priorities.”
We also describe this article a commentary and not a research study (lines 47 and 75).
- The authors should make the policy implications clearly about your study.
Response: We thank the reviewer for this suggestion. We have added a section before the conclusion describing the policy implications of a regional approach on lines 655-678.
“In providing a regional perspective to the issue of climate change-related health equity, the PARICHE approach and experience offers several implications for development and implementation of policies addressing climate change and health equity in other parts of the world. First, the principles of Community-Partnered Participatory Research are designed to eliminate the dominance of participants from HICs in climate health equity partnerships with participants from SIDS and LMICs. HICs have more resources but also bear more responsibility for human-induced climate change, while LMICs possess fewer resources but are more exposed and more vulnerable to climate change impacts [25]. Second, the goal of a regional approach is to empower those most affected by climate change to control the process of addressing health equity by dictating priorities and identifying interventions and strategies for implementation of policies, practices and programs that are both feasible and acceptable. Third, local empowerment should be a key priority in the implementation of international incentives and coordinating for scaling up climate adaptation. Efforts to understand means of effective climate risk reduction; assessing cascading, compounding, and transboundary risks; tracking adaptation progress; and financing adaptation such as those proposed by Magnan and colleagues [3], cannot succeed without simultaneously empowering communities in low resource settings like the SIDS and LMICs of the Pacific Region to engage in such activities as full partners.”
Reviewer 2 Report
Focusing on the regional approach to addressing, the authors describe an approach, which addresses climate-related health equity, by network of researchers, practitioners and policymakers dedicated to promoting climate-related health equity in Small Island Developing States and Low- and Middle-Income Countries in the Pacific. This article identifies three primary sets of needs related to developing regional capacity to address physical and mental health disparities through research, training and assistance in policy and practice implementation, with some conclusions proposed.
The topic is important, but it's hard to say the article is written in an academic format. Little analyses can be seen in the current manuscript, and the main work is introduction to the approaches. So, I suggest the authors to do enough researches according to the current topic.
Author Response
Focusing on the regional approach to addressing, the authors describe an approach, which addresses climate-related health equity, by network of researchers, practitioners and policymakers dedicated to promoting climate-related health equity in Small Island Developing States and Low- and Middle-Income Countries in the Pacific. This article identifies three primary sets of needs related to developing regional capacity to address physical and mental health disparities through research, training and assistance in policy and practice implementation, with some conclusions proposed.
The topic is important, but it's hard to say the article is written in an academic format. Little analyses can be seen in the current manuscript, and the main work is introduction to the approaches. So, I suggest the authors to do enough researches according to the current topic.
Response: We have added the following at the end of the introduction (lines 75-82) to clarify the aims of the article.
“In this commentary we describe one such regional approach comprised of partnerships involving researchers, practitioners and policymakers located along and within the Pacific Rim dedicated to addressing the health inequities that are either a consequence of climate change or have been exacerbated by climate change in the Pacific Region. With the overall goal of demonstrating the potential value of a regional approach, the aims of this commentary are threefold: 1) to describe the impacts of climate change on health equity in the Pacific region; 2) to outline the priorities for addressing these impacts on health equity; and 3) to describe the efforts of the Pacific Rim Climate Health Equity (PARICHE) network to conduct research and capacity building in accordance with these priorities.”
We also describe this article a commentary and not a research study (lines 47 and 75).
Reviewer 3 Report
I really like the paper but it lacks detail in some places. By including this, the paper will be stronger.
The paper makes sense to me and resonates with my work on climate change and health and wellbeing and equity.
One thing the paper lacks is what have been the key impacts of the PARICHE work to date? yes there are examples of work but has this made a difference? i don't see that in the conclusion of the paper.
I have some other minor comments below:
Line 77 on - this is a very long sentance and does not make much sense!
line 82 - define Atoll countries
line 104 - Another health-related consequence of climate change is the physical and mental trauma - this is all very clinically focussed. the work in Wales shows significant impacts on mental wellbeing and the protective factors of this and the population groups affected https://phw.nhs.wales/news/new-resource-highlights-health-impacts-of-climate-change/climate-change-infographics-english/
line 150 - PARICHE ....is an exemplar of a regional approach to climate health equity. Reference to this? who says this?
line 157 - representatives from SIDS/LMIC government agencies and international organizations. who exactly? please provide examples.
171 - uncertainty - again can you refer to wellbeing also? uncertainty and anxiety are key components of mental wellbeing
line 198 - the examples provided are from almost 10 years ago - are there no more recent examples? also could these (or one of these) examples be presented in a box as a short case study?
line 206 - who was at the round table? how was SOLAR constructed and did you test it?
Author Response
I really like the paper but it lacks detail in some places. By including this, the paper will be stronger.
The paper makes sense to me and resonates with my work on climate change and health and wellbeing and equity.
Response: We thank the reviewer for this comment.
One thing the paper lacks is what have been the key impacts of the PARICHE work to date? yes there are examples of work but has this made a difference? i don't see that in the conclusion of the paper.
Response: We thank the reviewer for this recommendation. We have provided more detail on the impacts of our work to date including 1) the application of the SOLAR intervention in Tuvalu (O’Donnell, Forbes and Gibson), 2) the training of Filipino educators and Peace Corps workers in evidence-based mental health interventions (Villaverde and Wong), 3) and the outcomes of the diabetes research conducted in Samoa (Hawley). We have also added an example of work with Marshallese Islanders (Saunders).
I have some other minor comments below:
Line 77 on - this is a very long sentance and does not make much sense!
We have revised this sentence as follows:
“One example of the need for such an approach is the Small Island Developing States (SIDS) and territories and larger LMICs within the Pacific region that are disproportionately impacted by climate change relative to other regions in the world. In this region, climate change has potentially devastating implications for Indigenous peoples and their traditional lifeways [2,4]”
line 82 - define Atoll countries
Response: Atoll countries are Small Island Developing States that are located in atolls (a ring-shaped coral reef, island, or series of islets. We provided this definition on lines 91-92 of the revised manuscript.
line 104 - Another health-related consequence of climate change is the physical and mental trauma - this is all very clinically focussed. the work in Wales shows significant impacts on mental wellbeing and the protective factors of this and the population groups affected https://phw.nhs.wales/news/new-resource-highlights-health-impacts-of-climate-change/climate-change-infographics-english/
Response: We thank the reviewer for sharing this infographic with us. We have inserted the following on lines 147-156.
“For instance, many of same the impacts of climate change on well-being and the factors that mitigate these impacts can be found in other parts of the world such as Wales [17]. Like Wales, the Pacific region is confronting increased food insecurity and SLR. However, unlike Wales, SLR is threatening to immerse entire nation states on low-lying islands and Pacific SIDS have fewer resources to develop the infrastructure necessary to mitigate the impacts of SLR. Moreover, community responses to dealing with food insecurity in the Pacific region (e.g., traditional kin-based networks of food exchange and social support) are very different from those in operation in Wales (government assistance, non-government organization food distribution programs).”
We have also included the reference used to develop the infographic. Furthermore, we cite on lines 346-352 examples of efforts to address well-being that are not clinically focused
“There is also a need to ensure access to healthy, culturally relevant foods and sustainable agriculture resistant to climate change and reduce the risk of malnutrition, which is related to both infectious and non-infectious diseases [32]. Over avenues for promotion of general well-being in the region include engagement in indigenous “Pasifika arts [19], planned relocation of residents from SIDS at risk of SLR [33], and partnerships for implementation of community-based health promotion programs [34].”
line 150 - PARICHE ....is an exemplar of a regional approach to climate health equity. Reference to this? who says this?
Response: We have clarified the intent of the sentence to state that PARICHE is a potential exemplar of a regional approach. The aim of the commentary was to argue for a regional approach
line 157 - representatives from SIDS/LMIC government agencies and international organizations. who exactly? please provide examples.
Response: we have provided the following examples on lines 178-180 of the revised manuscript: Asian Disaster Preparedness Center, International Organization for Migration (Marshall Islands, Federated States of Micronesia, Western Pacific), and the Provincial Government of New Ireland, Papua New Guinea.
171 - uncertainty - again can you refer to wellbeing also? uncertainty and anxiety are key components of mental wellbeing
Response: We have included well-being in the sentence.
line 198 - the examples provided are from almost 10 years ago - are there no more recent examples? also could these (or one of these) examples be presented in a box as a short case study?
Response: We provide more detail on more recent collaborations in three boxes as suggested by the reviewer: the evaluation of SOLAR to address the mental health impacts of Tropical Cyclone Pam on survivors in Tuvalu; research to address risk factors for noncommunicable disease in Samoa; and responses to the COVID-19 pandemic in the Marshall Islands and elsewhere in the Pacific.
line 206 - who was at the round table? how was SOLAR constructed and did you test it?
Response: We describe in the revised manuscript that PARICHE members O’Donnell, Forbes and Ugsang were part of the roundtable. We also note in Box 1 that SOLAR was initially evaluated with bushfire survivors in Australia and subsequently adapted with assistance of the Tuvalu partners.
Round 2
Reviewer 2 Report
The authors have revise the manuscript, and I think the current copy can be accepted as a communication with minor revision.
I suggest the authors revise Fig.1. The current figure can only indicate geography information on some countries, and the authors can process it to show or highlight some insights.
Author Response
We thank the reviewer for her/his suggestion. We have identified a map of the region that is more in line with your suggestion. We hope you find it satisfactory.
